# Dietary lipids shape cytokine and leptin profiles in obesity-metabolic syndrome implications: A cross-sectional study

Sakawrut Poosri[1], Karani Santhanakrishnan Vimaleswaran[2,3], Pattaneeya Prangthip[1]*

**1** Faculty of Tropical Medicine, Department of Tropical Nutrition and Food Science, Mahidol University, Bangkok, Thailand, **2** Department of Food and Nutritional Sciences, Institute for Cardiovascular and Metabolic Research (ICMR), Hugh Sinclair Unit of Human Nutrition, University of Reading, Reading, United Kingdom, **3** Institute for Food, Nutrition and Health (IFNH), University of Reading, Reading, United Kingdom

* pattaneeya.pra@mahidol.ac.th

**Data Availability Statement:** All relevant data are within the paper and its Supporting Information files.

## Abstract

### Background

Obesity, characterized by chronic energy imbalance and excessive adiposity, is a key component of metabolic syndrome and is associated with low-grade inflammation and altered adipokine secretion. This study aimed to evaluate the association between dietary fat consumption and its influence on interleukin (IL) and leptin levels in participants with obesity.

### Methods

Using the Asian obesity classification criteria, a cross-sectional study was conducted on 384 adults (18–59 years). Anthropometric measurements by bioelectrical impedance analyzer (BIA), blood biochemistry by colorimetric assay, inflammatory markers and hormones by ELISA test, and dietary intake were assessed by Semi-FFQ.

### Results

Obesity prevalence was 26.1% and 73.90% in males and females, respectively. Participants with obesity exhibited significantly higher inflammatory and hormonal marker levels. Positive correlations were observed between blood lipid, glucose, and tumor necrosis factor-α, IL-6, and leptin levels. Energy, carbohydrate, and sugar intake were positively correlated with leptin levels. High saturated fat intake was associated with increased IL-6 levels (odds ratio = 2.03, 95% confidence interval [CI] = 1.00–4.11, p < 0.047), whereas high total fat intake elevated leptin levels by 2.14-fold (95% CI = 1.12–4.10, p < 0.021) in participants with obesity.

### Conclusions

This study demonstrates significant associations between dietary fat composition, inflammatory markers, and leptin levels in individuals with obesity. These findings suggest that modulating dietary fat intake can be a potential strategy for mitigating obesity-related inflammation and leptin resistance, highlighting the need for targeted nutritional interventions in obesity and metabolic syndrome management.

**Funding:** This research paper is supported by Specific League Funds from Mahidol University awarded to PP.

**Competing interests:** The authors have declared that no competing interests exist.

**Abbreviations:** BMI, Body mass index; HDL, High-density lipoprotein; LDL, Low-density lipoprotein; TG, Triglyceride; SFAs, Saturated fatty acids; MUFAs, Monounsaturated fatty acids; IL-6, Interleukin-6; TNF-α, Tumor necrosis factor alpha; FFM, Fat-free mass; BMR, Basal metabolic rate; MUAC, Middle upper arm circumference; HOMA-IR, Homeostatic model assessment for insulin resistance.

# Background

Obesity is a significant risk factor for metabolic syndrome, a cluster of conditions including diabetes, cardiovascular diseases (CVDs), dyslipidemia, and insulin resistance [1]. The Global Burden of Disease Study (2017) indicated that the prevalence of obesity has dramatically increased over the past few decades, affecting millions globally [2]. Specifically, the NCD Risk Factor Collaboration (2016) reported that from 1975 to 2016, the global prevalence of obesity nearly tripled, with over 650 million adults classified as obese, underscoring the urgency of addressing this epidemic [3]. Furthermore, a systematic review and meta-analysis highlighted that obesity rates are projected to continue rising, with an anticipated prevalence of 1.12 billion adults expected to be obese by 2030 [4]. Complex factors of genetic susceptibility, sedentary lifestyle, and high-calorie diet consumption comprise the etiology of obesity and metabolic syndrome, resulting in chronic energy imbalance and excessive body fat accumulation [5,6]. The global rise in obesity is closely linked to changes in dietary patterns, particularly the increased consumption of high-fat diets. Lipids play a crucial role in the development of obesity and related metabolic disorders.

Obesity and metabolic syndrome are characterized by chronic low-grade inflammation, significantly impacting adipose tissue functionality and associated with an excessive increase in visceral adipose tissues [7]. White adipose tissues secrete various pro- and anti-inflammatory substances, including cytokines, such as interleukin (IL-6) and tumor necrosis factor-α (TNF-α), and adipokines, such as leptin and adiponectin [8]. These inflammatory factors have been linked to the development and progression of insulin resistance, type 2 diabetes (T2D), and cardiovascular complications in metabolic syndrome [9]. Dysregulated lipid metabolism contributes not only to excess body fat but also to inflammatory processes that involve cytokines such as interleukin-6 (IL-6) and tumor necrosis factor-alpha (TNF-α), as well as adipokines like leptin. These molecules are key mediators in the pathogenesis of obesity-related complications, including insulin resistance and cardiovascular disease [10].

Leptin, primarily produced by adipocytes in white adipose tissues, plays a crucial role in hunger, body weight, and glucose metabolism regulation [11]. In both males and females, leptin levels are positively correlated with adiposity. However, obesity and metabolic syndrome have been associated with leptin resistance, partly because of abnormalities in leptin receptor signaling and impaired leptin transport across the blood–brain barrier [12]. In obese individuals, leptin levels are significantly elevated due to the increased adipose tissue mass. Despite these high levels, the body becomes resistant to leptin's effects, leading to continued overeating and reduced energy expenditure. This leptin resistance is a key factor in the pathogenesis of obesity, as it undermines the body's ability to regulate weight effectively [13]. Leptin not only regulates energy balance but also functions as a pro-inflammatory mediator. Elevated leptin levels in obesity are associated with increased production of inflammatory cytokines, such as TNF-α and IL-6, which further contribute to chronic low-grade inflammation seen in obesity and metabolic syndrome [14]. This inflammatory state is linked to insulin resistance, cardiovascular diseases, and other obesity-related complications.

Major lifestyle factors contributing to the onset and progression of obesity, metabolic syndrome, and their comorbidities include diet and physical activity. Dietary components, particularly dietary fat, can influence inflammation and disease risk [15]. The lipid content of the diet is strongly correlated with obesity and metabolic disturbance onset and duration [16]. High fat diets (HFDs) have been shown to exacerbate the obesity epidemic and the emergence of associated metabolic diseases, including T2D [17]. Studies have indicated that HFD consumption can result in food-induced obesity and metabolic abnormalities that mimic the metabolic syndrome in humans [18]. Increased saturated fat intake has been linked to elevated

levels of inflammatory indicators and insulin resistance, primarily through toll-like receptor 4 pathway activation [19].

Furthermore, diet influences serum leptin levels, inflammation, and insulin sensitivity. In humans and rats, energy restriction and fasting have been shown to lower blood leptin levels and improve insulin sensitivity [20]. Preliminary research has suggested that the fatty acid composition of the diet can influence serum leptin levels and metabolic parameters [21]. Studies in rats and human cell lines have shown varying effects of different fatty acid types (n-6, n-3 polyunsaturated, monounsaturated, and saturated) on leptin gene expression, serum levels, and insulin signaling [22,23]. Despite extensive research on obesity and metabolic syndrome, the relationship between dietary fat content, serum leptin levels, and inflammation markers, particularly in individuals with obesity with or at risk of metabolic syndrome, remains incompletely understood. While extensive research has been conducted globally on the relationship between diet, lipid metabolism, and obesity, there is a lack of studies that explore these factors within specific ethnic and geographical contexts. The social, cultural, and environmental factors unique to particular populations can significantly influence dietary patterns and their metabolic effects [24]. Understanding these contextual variables is essential to ensuring the external validity of any findings and to allow for the generalization of results across diverse populations. This study addresses the gap in understanding the relationships between dietary fat types, inflammation markers, and leptin levels in individuals with obesity, particularly in non-Western populations. We aim to provide a comprehensive view of these factors in Thai adults with obesity, combining detailed dietary assessment with biomarker analysis. This novel approach in selecting specific population has the potential to reveal mechanisms of obesity-related metabolic disorders and inform culturally appropriate strategies for metabolic syndrome management in Southeast Asia.

## Methods

### Study design and participants

This cross-sectional study was conducted between July and October 2023 in the Laksi District of Bangkok, Thailand. This study included 384 adults aged 18–59 years who were recruited on the basis of predetermined inclusion and exclusion criteria. The following were the inclusion criteria: literacy, body mass index (BMI) $\geq 18.5$ kg/m$^2$ to 40 kg/m$^2$, and willingness to participate. The following encompassed the exclusion criteria: pregnancy, lactation, severe chronic illnesses, metabolic disorders affecting nutritional needs, eating disorders, recent significant weight loss, and cannot provide blood samples.

### Procedures

The study protocol was approved by the Ethics Committee of the Faculty of Tropical Medicine, Mahidol University of Thailand (EC approval number: MUTM 2023-042-01) and adhered to the Declaration of Helsinki guidelines. All participants provided informed consent before study participation. Thai clinical trials no. is TCTR20240723011.

The study collected data from August 1st to September 30th, 2023. During this period, public relations efforts were carried out in the Laksi community to encourage participant registration. These activities included displaying billboards, sharing project infographics, and distributing posters to inform and engage potential participants. The researcher contacted participants via telephone to screen participants who meet the inclusion/exclusions criteria and recruited to the study.

One day before the study, the researcher contacted participants to remind them to fast from food and water for 12 hours. On the day of the study, the researcher provided a verbal

explanation of the study details and participant information to ensure everyone understood the procedures and purpose of the research. The participant read patient information sheet and written inform consent form. Then, following the 4 stations to complete the study process, 1st Station: Volunteers completed the questionnaire by trained interviewers. Station 2: Measurement of the vital signs; such height, weight, and blood pressure as well as drawing two tablespoons of blood under a certified nurse's observation. Station 3: Measurement of anthropometry by bioelectrical impedance analyzer (BIA), and body composition. Station 4: Assessment of dietary consumption by Semi-FFQ with an interviewer.

## Measurement

**Anthropometric and body composition measurements.** Anthropometric parameters (height, weight, and waist and hip circumferences) were measured using standardized techniques following the International Biological Program criteria. BMI was calculated as weight (kg) divided by height squared ($m^2$). The waist to hip ratio was determined. Body composition was assessed using a bioelectrical impedance analyzer (BIA) (TANITA®-SC330, Tanita Corporation, Tokyo, Japan) to evaluate muscle-mass, fat-mass, and metabolic components.

**Biochemical analyses.** After a 12-h fast, blood samples were collected for glucose, lipid profile, and inflammatory marker and hormone analyses. Serum was separated and stored at −80˚C until analysis.

**Lipid profiles:** total cholesterol, high-density lipoprotein (HDL), and triglyceride levels were measured using enzymatic methods (Stanbio Cholesterol LiquiColor® Test Kit, Boerne, Texas, USA). Low-density lipoprotein (LDL)-cholesterol levels were calculated using the Friedewald formula. Fasting glucose levels were determined using the glucose oxidase method (Glucose LiquiColor® Test Kit, Stanbio Laboratory, Boerne, Texas, USA). Homeostatic model assessment for insulin resistance (HOMA-IR) was computed as (fasting glucose [mg/dL] × fasting serum insulin [mU/L]/405) [25].

**Inflammatory markers and hormones:** Myeloperoxidase, IL-6, and TNF-α were quantified using enzyme-linked immunosorbent assay (ELISA) kits from MyBioSource Inc. (San Diego, CA, USA). Leptin and insulin levels were measured using ELISA kits from Sigma-Aldrich (St. Louis, MO, USA). All assays were performed according to manufacturers' protocols. Cutoff values indicating high risk for metabolic diseases and inflammation-related complications were determined based on previous research in obesity and metabolic syndrome. These thresholds predict increased risk for metabolic disorders (insulin resistance, cardiovascular diseases) and systemic inflammation, with established clinical significance ($p < 0.05$) in multiple cohort studies. The following values were defined as high-risk: TNF-α, >29 pg/mL 0; IL-6, >30 pg/mL; myeloperoxidase, >87.8 ng/mL; leptin, >5 ng/mL; and insulin, >12 uIU/mL [26–29].

**Dietary assessment.** The semi-FFQ was designed to assess dietary intake over a 1-month period. Dietary intake was assessed using a semi-quantitative food frequency questionnaire (FFQ) containing 75 commonly consumed dishes and beverages in Bangkok, developed in a previous study [30]. The FFQ categorized food items into 12 groups on the basis of nutrient composition and obesity risk factors. The semi-FFQ, initially developed for the SI-Health study. A 5-level scale is used for serving sizes based on the 'household unit' of the Thai food-based dietary guidelines. The questionnaire provides information on food consumption frequency and portion sizes, with nutrient profiling to classify foods into three risk levels for non-communicable diseases (NCDs). Energy and nutrient content were calculated using the INMUCAL-Nutrients V.4.0 software (Institute of Nutrition, Mahidol University, Nakhon Pathom, Thailand). High saturated fat consumption was defined as >22 grams, total fat > 70

grams, and high sugar content > 50 grams, based on the World Health Organization guidelines and previous studies [31].

## Statistical analysis

The sample size calculation was conducted through two approaches to ensure adequate statistical power. First, for detecting genetic associations, a power analysis was performed with parameters of moderate effect size (r = 0.3), power (1-β) = 0.80, and significance level (α) = 0.05 for correlation analysis, indicating a minimum required sample of 374 participants. Second, for population proportion estimation, we used the formula n = [$Z^2$p(1-p)]/$d^2$, where Z = 1.96 (95% confidence level), p = 0.424 (42.4% obesity prevalence in Thai population, 2021), and d = 0.05 (5% margin of error), which also yielded a minimum sample size of 374. To account for potential 10% data loss, we aimed to recruit 412 participants. The final study included 384 participants, exceeding the minimum required sample size of 374, thus ensuring adequate statistical power for both genetic association analysis and population proportion estimation.

Data were analyzed using Statistical Package for the Social Sciences (version 23, IBM, Armonk, NY, USA). Continuous variables were expressed as means ± standard deviations and categorical variables as percentages. Obesity was defined according to Asian classification (BMI $\geq$ 25 kg/$m^2$). The associations between dietary lipids and cytokine profiles were assessed using multivariable logistic regression models, adjusting for potential confounders including age, gender, and physical activity. Relationships between dietary factors and inflammatory markers were analyzed using Chi-square tests for categorical variables and Student's t-tests for continuous variables. Linear regression analysis, adjusted for gender, was performed to examine associations between metabolic parameters. Correlations between variables were evaluated using Pearson correlation coefficients. Statistical significance was set at $p < 0.05$.

## Results

### Participant characteristics

The characteristics of the study population are presented in Table 1. The study included a total of 384 adults, comprising 299 females and 85 males. Females (73.90%) had higher obesity prevalence than males (26.1%). Most participants with obesity were aged 46–59 years (46.23%), followed by 18–35 years (31.16%). Educational attainment significantly differed between the non-obese and obese groups, with participants with obesity being more likely to have a bachelor's degree (32.66%) than those who attained high school education (26.13%). Occupational distribution did not significantly differ between the groups. Both groups reported low rates of smoking and regular exercise.

### Anthropometric parameters

Table 2 provides a comprehensive overview of the anthropometric parameters and the distribution of participants across the non-obese and obese categories. In the obese group, all anthropometric parameters after adjusted gender were significantly higher ($p < 0.001$), except for age and height. Notably, the average waist circumference in the obese group exceeded 90 cm, correlating with increased CVD risk.

### Nutritional intake

In the obese group, energy, carbohydrate, protein, sugar, total fat, and saturated fatty acid (SFA) intakes were significantly higher ($p < 0.05$). Additionally, in the obese group, cholesterol

**Table 1. Characteristics of the study population.**

| Variables | Non-obese (%) (n = 185) | Obese (%) (n = 199) | P value |
|---|---|---|---|
| **Gender** | | | 0.051 |
| • Male | 33 (17.8) | 52 (26.1) | |
| • Female | 152 (82.2) | 147 (73.90) | |
| **Age (Years)** | | | 0.093 |
| • 18–35 | 73 (39.46) | 62 (31.16) | |
| • 36–45 | 37 (20.00) | 45 (22.61) | |
| • 46–59 | 75 (40.54) | 92 (46.23) | |
| **Education** | | | **0.01** |
| • Uneducated | 3 (1.62) | 0 (0) | |
| • Primary school | 14 (7.57) | 31 (15.58) | |
| • High School | 38 (20.54) | 52 (26.13) | |
| • Vocational college | 22 (11.89) | 25 (12.56) | |
| • Bachelor degree | 76 (41.08) | 65 (32.66) | |
| • Higher Bachelor degree | 32 (17.30) | 26 (13.07) | |
| **Occupation** | | | 0.145 |
| • Unemployed | 21 (11.35) | 22 (11.06) | |
| • Personal business | 31 (16.67) | 41 (20.60) | |
| • Employee/House work | 44 (23.78) | 63 (31.66) | |
| • Office | 6 (3.24) | 7 (3.52) | |
| • Government service | 55 (29.73) | 46 (23.12) | |
| • Others/student | 28 (15.14) | 20 (10.05) | |
| **Smoking** | | | 0.507 |
| • No | 173 (93.51) | 184 (92.46) | |
| • Yes | 12 (6.49) | 15 (7.54) | |
| **Exercises (> 150 min/week)** | | | 0.849 |
| • No | 112 (60.54) | 122 (61.31) | |
| • Yes | 73 (39.46) | 77 (38.69) | |
| **Physical activity** | | | 0.447 |
| • Physical inactivity | 113 (61.08) | 109 (54.77) | |
| • Moderate physical activity | 55 (29.73) | 80 (40.20) | |
| • High activity | 17 (9.19) | 10 (5.03) | |

P values are calculated using the Chi-square test.

* P values of <0.05 are considered statistically significant.

and sodium intakes showed an increasing trend although not statistically significant. Table 3 presents a detailed comparison of nutritional intake between the non-obese and obese groups.

## Blood inflammation, hormonal, and biochemical markers

Fig 1 presents a comparison of blood inflammation markers, hormonal levels, and blood biochemistry parameters between non-obese and obese groups. In the obese group, inflammatory marker (TNF-α and IL-6), myeloperoxidase, leptin, and insulin levels were significantly elevated ($p < 0.05$). Moreover, blood biochemistry parameters excluding total cholesterol were significantly higher in the obese group ($p < 0.05$) and HDL level was higher in non-obese group significantly ($p < 0.05$).

**Table 2. Anthropometric parameters and the number of participants with and without obesity.**

| Variables | Non-obese (n = 185) | Obese (n = 199) | P value | |
|---|---|---|---|---|
| | Mean ± SD | Mean ± SD | | P value[a] |
| Age (years) | 39.81 ± 12.00 | 42.04 ± 11.37 | 0.063 | 0.443 |
| Hight (cm) | 159.42 ± 7.29 | 160.00±8.34 | 0.478 | 0.368 |
| Weight (kg) | 56.04±6.70 | 76.89±16.08 | **0.000*** | **0.000*** |
| Fat (%) | 27.36±6.02 | 37.78±7.94 | **0.000*** | **0.000*** |
| Fat-mass (kg) | 15.31±3.95 | 29.40±11.34 | **0.000*** | **0.000*** |
| Fat-free mass (kg) | 40.64±6.34 | 47.20±9.78 | **0.000*** | **0.001*** |
| Muscle-mass (kg) | 38.37±6.03 | 44.64±9.46 | **0.000*** | **0.000*** |
| Basal Metabolic rate (kcal) | 1189.72±166.83 | 1435.78±278.19 | **0.000*** | **0.000*** |
| Metabolic age (years) | 33.46±9.70 | 55.50±10.45 | **0.000*** | **0.000*** |
| Visceral fat rating | 5.19±2.39 | 10.63±3.86 | **0.000*** | **0.000*** |
| Body mass index (kg/m$^2$) | 22.02±1.68 | 29.87±5.10 | **0.000*** | **0.000*** |
| Waist circumference (cm) | 76.05±9.69 | 94.54±12.32 | **0.000*** | **0.000*** |
| Hip circumference (cm) | 94.96±4.77 | 108.02±9.59 | **0.000*** | **0.000*** |
| Waist to Hip ratio | 0.79±0.09 | 0.87±0.07 | **0.000*** | **0.000*** |
| Neck circumference (cm) | 32.67±3.23 | 37.21±4.06 | **0.000*** | **0.000*** |
| Middle arm circumference (cm) | 26.31±2.52 | 32.21±3.58 | **0.000*** | **0.000*** |
| Blood pressure (mmHg) | | | | |
| Systolic | 119.26±16.59 | 133.92±18.20 | **0.000*** | **0.000*** |
| Diastolic | 75.66±10.54 | 83.86±12.58 | **0.000*** | **0.000*** |

P values are calculated using Student's t-test. Data are presented as means ± standard deviations (SDs).

P values[a] are calculated using Linear regression adjusted by gender

* P values of <0.05 are considered statistically significant.

## Correlations between markers and nutritional intake

Table 4. presents correlation between blood inflammation markers, hormonal and nutrition composition, providing insights into how dietary factors may influence various physiological markers in the context of obesity. Myeloperoxidase levels were positively correlated with sugar and total cholesterol intakes (r = 0.151 and 0.121, respectively; $p < 0.05$). Leptin levels were positively correlated with energy, carbohydrate, and sugar intakes (r = 0.124, 0.140, and 0.110, respectively; $p < 0.05$). Table 5 showed TNF-α levels was positively correlated with blood glucose, total cholesterol, triglyceride, and LDL levels (r = 0.141, 0.122, 0.215, and 0.142, respectively; $p < 0.05$). Leptin level was positive correlated with blood glucose, total cholesterol, triglyceride, and LDL levels (r = 0.154, 0.102, 0.155, and 0.139, respectively, $p < 0.05$). TNF-α, IL-6, leptin, insulin, and HOMA-IR were negatively correlated with HDL levels (r = −0.229, −0.143, −0.253, −0.230, and −0.246, respectively; $p < 0.001$). IL-6 levels were positively correlated with blood glucose, triglyceride, and LDL levels (r = 0.115, 0.145, and 0.103, respectively; $p < 0.05$).

## Associations in participants with obesity

S1–S3 Tables express interactions between blood inflammation and hormonal markers in relation to sugar, saturated fat, and total fat intake in the obese group. After adjusting for age and

**Table 3. Nutrition consumption of the non-obese and obese groups.**

| Variables | Non-obese (n = 185) | Obese (n = 199) | P value |
|---|---|---|---|
| | Mean ± SD | Mean ± SD | |
| **Nutrients** | | | |
| Energy (kcal/day) | 2249.64±626.47 | 2565.93±1044.46 | **0.001** |
| Carbohydrate (g) | 262.18±84.03 | 301.82±117.02 | **0.000** |
| Sugar (g) | 65.14±33.02 | 75.58±44.39 | **0.011** |
| Protein (g) | 102.08±32.38 | 115.92±55.87 | **0.004** |
| Total Fat (g) | 88.07±29.37 | 99.75±50.42 | **0.007** |
| SATFAT (g) | 26.49±8.38 | 29.92±15.10 | **0.008** |
| Cholesterol (g) | 461.86±193.17 | 511.21±294.08 | 0.059 |
| Sodium (mg) | 3310.96±2271.57 | 3634.27±2510.43 | 0.197 |
| Carbohydrate (%) | 46.52±7.07 | 47.80±8.45 | 0.164 |
| Protein (%) | 18.33±3.08 | 18.01±3.20 | 0.341 |
| Fat (%) | 35.15±5.40 | 34.18±6.64 | 0.128 |

P values are calculated using Student's t-test. Data are presented as means ± SDs.

* P values of <0.05 are considered statistically significant.

gender, high saturated fat intake was associated with increased IL-6 levels (odds ratio [OR] = 2.03, 95% confidence interval [CI] = 1.00–4.11, $p < 0.047$) compared with low saturated fat intake. High total fat intake was associated with increased leptin levels (OR = 2.14, 95% CI = 1.12–3.38, $p < 0.021$) compared with low fat intake. Although high sugar intake showed a trend toward increased leptin levels (OR = 2.12), this association was not statistically significant as shown in S1 Table.

## Discussion

Individuals with BMIs of 25 kg/m$^2$ or more are considered obese, which is a major risk factor for metabolic disorders, including T2D and CVDs, and is independently linked to greater mortality. Our study, which included 384 adults, noted that females (73.90%) had higher obesity prevalence than males (26.1%), with most participants with obesity aged 46–59 years (46.23%). To address the higher prevalence of obesity among women, both globally and specifically in Asia, several factors need to be considered, including biological, sociocultural, and economic influences. Globally, women tend to have higher obesity rates than men due to

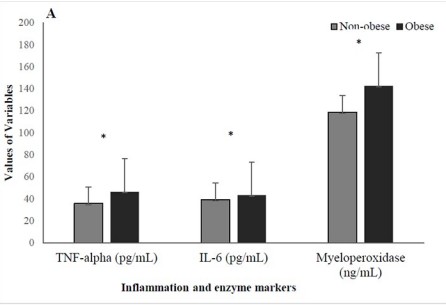 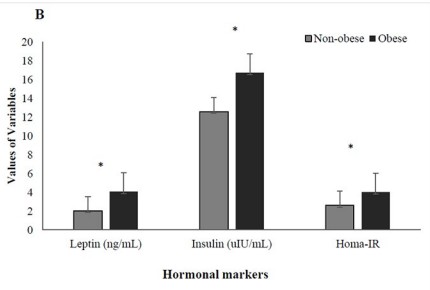 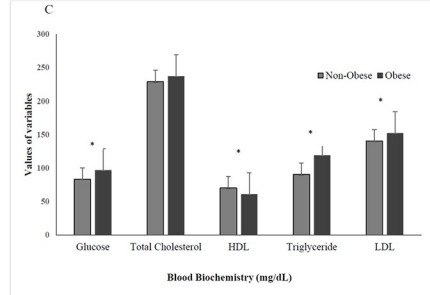

**Fig 1.** A: Blood inflammation and enzyme markers of non-obese and obese group B: Hormonal marker of non-obese and obese group C: Blood biochemistry of non-obese and obese group. P value were calculated by Student's t test compared between both groups. * is statistically significant at *p value* <0.05.

**Table 4. Correlation between blood inflammation markers, hormonal, and nutrition composition.**

| | | Energy | CHO | sugar | Protein | Total fat | SFAs | Cholesterol | %CHO | %Protein | %FAT |
|---|---|---|---|---|---|---|---|---|---|---|---|
| **TNF-α** | Pearson Correlation | 0.025 | 0.053 | 0.04 | 0.02 | -0.009 | -0.018 | -0.056 | 0.091 | -0.024 | -0.105 |
| | *p value* | 0.631 | 0.311 | 0.445 | 0.698 | 0.868 | 0.732 | 0.283 | 0.083 | 0.648 | 0.146 |
| **IL-6** | Pearson Correlation | -0.027 | -0.008 | 0.019 | -0.037 | -0.036 | -0.043 | -0.054 | 0.028 | -0.007 | -0.033 |
| | *p value* | 0.604 | 0.878 | 0.712 | 0.48 | 0.5 | 0.417 | 0.302 | 0.591 | 0.896 | 0.532 |
| **Myeloperoxidase** | Pearson Correlation | 0.025 | 0.032 | .151** | -0.024 | 0.035 | 0.054 | .121* | 0.012 | -0.088 | 0.03 |
| | *p value* | 0.637 | 0.541 | **0.004** | 0.642 | 0.507 | 0.306 | **0.021** | 0.822 | 0.095 | 0.567 |
| **Leptin** | Pearson Correlation | .124* | .140** | .110* | 0.078 | 0.097 | 0.095 | 0.031 | 0.082 | -0.081 | -0.064 |
| | *p value* | **0.018** | **0.008** | **0.036** | 0.14 | 0.064 | 0.072 | 0.559 | 0.119 | 0.124 | 0.227 |
| **Insulin** | Pearson Correlation | 0.007 | 0.053 | 0.016 | -0.018 | -0.032 | -0.047 | -0.051 | 0.092 | -0.042 | -0.097 |
| | *p value* | 0.893 | 0.315 | 0.765 | 0.731 | 0.546 | 0.37 | 0.336 | 0.08 | 0.429 | 0.065 |

**. Correlation is significant at *p values* of <0.01.

*. Correlation is significant at *p values* of <0.05.

Abbreviations: CHO as a carbohydrate, SFAs as a saturated fatty acid.

physiological factors such as differences in fat distribution and hormonal that predispose them to higher fat accumulation [32]. Sociocultural norms, such as expectations around body image, diet, and physical activity, also disproportionately affect women, with many cultures imposing greater limitations on women's mobility and exercise opportunities [33]. In Asia, rapid urbanization and economic transitions have resulted in lifestyle changes, such as a shift toward higher consumption of processed and energy-dense foods combined with more sedentary behaviors, which disproportionately affect women [34]. The combination of these factors can lead to an increase in the obesity prevalence among women in these regions. Furthermore, the nutrition transition in many developing Asian countries has significantly contributed to the rising obesity rates in women, particularly as traditional diets give way to more

**Table 5. Correlation between blood inflammation markers, hormonal, and blood biochemistry.**

| | | Blglucose | Blcholes | BLHDL | BLTG | BLLDL |
|---|---|---|---|---|---|---|
| TNF-α | Pearson Correlation | .141** | .122* | -.229** | .215** | .142** |
| | Sig. (2-tailed) | **.006** | **.018** | **.000** | **.000** | **.006** |
| IL-6 | Pearson Correlation | .115* | .092 | -.143** | .145** | .103* |
| | Sig. (2-tailed) | **.025** | .073 | **.005** | **.005** | **.044** |
| Myeloperoxidase | Pearson Correlation | .217** | -.005 | -.056 | .093 | -.011 |
| | Sig. (2-tailed) | **.000** | .929 | .275 | .071 | .835 |
| Leptin | Pearson Correlation | .154** | .102* | -.253** | .155** | .139** |
| | Sig. (2-tailed) | **.003** | **.048** | **.000** | **.002** | **.006** |
| Insulin | Pearson Correlation | .357** | .031 | -.230** | .211** | .046 |
| | Sig. (2-tailed) | **.000** | .547 | **.000** | **.000** | .372 |
| HomaIR | Pearson Correlation | .756** | .056 | -.246** | .345** | .048 |
| | Sig. (2-tailed) | **.000** | .276 | **.000** | **.000** | .355 |

*. Correlation is significant at the 0.05 level (2-tailed).

**. Correlation is significant at the 0.01 level (2-tailed).

Abbreviations: Blglucose as a Blood glucose, Blccholes as a Blood cholesterol, BLHDL as a Blood HDL, BLTG as a Blood triglyceride, BLLDL as a Blood LDL.

Westernized eating patterns [35]. This finding aligns with the understanding that obesity results from a imbalance between energy intake and expenditure, influenced by genetics, nutrition, and environmental factors. An individual's diet and activity level are two key lifestyle factors that influence how obesity and its comorbidities start and worsen [36].

In participants with obesity, we observed significantly elevated inflammatory marker (TNF-α and IL-6), myeloperoxidase, leptin, and insulin levels ($p < 0.05$). This finding supports the concept that obesity is characterized by increasing inflammation and immune cell infiltration into adipocytes [37]. The positive correlations observed between the levels of these markers and blood glucose, lipids, and LDL levels further emphasize the relationship between obesity, inflammation, and metabolic disorders [38]. Our negative correlations between HDL and inflammatory markers (TNF-α, IL-6), leptin, and insulin also suggest a relationship where inflammation may impair HDL synthesis and increase its catabolism. This could create a cycle where reduced HDL levels further compromise anti-inflammatory protection in obesity. Obesity is associated with chronic low-grade inflammation, largely due to the excessive expansion of adipose tissue. As adipocytes expand, immune cells, particularly macrophages, infiltrate the adipose tissue and secrete pro-inflammatory cytokines such as TNF-α and IL-6, which play a key role in the development of insulin resistance. TNF-α disrupts insulin signaling through the activation of NF-κB and other inflammatory pathways, impairing glucose uptake in peripheral tissues like muscle and liver [39]. Similarly, IL-6 promotes hepatic glucose production and exacerbates insulin resistance [40]. Leptin, a hormone predominantly secreted by adipocytes, is also elevated in obesity, but its normal regulatory function on appetite and energy expenditure becomes impaired, leading to leptin resistance [41]. High circulating triglycerides have been implicated in disrupting leptin transport across the blood-brain barrier, reducing leptin's ability to signal satiety, which exacerbates overeating and weight gain [42]. This condition of leptin resistance is commonly observed in obesity and contributes to its progression [41].

Our nutritional intake analysis revealed significantly higher energy, carbohydrate, protein, sugar, total fat, and SFA consumption in the obese group ($p < 0.05$). This finding aligns with previous findings that dietary fat is a crucial factor in obesity development. Certain dietary components including dietary fat can influence inflammation in humans. The fatty acids that make up dietary fat include polyunsaturated fatty acids of the omega (n) 6- and n3-family, conjugated linoleic acid, monounsaturated fatty acids (MUFAs), and saturated and trans fatty acids. One possible mechanism by which dietary fat modulates inflammation is through eicosanoid compound metabolism or it may operate by activating gene expression to regulate membrane and cytosolic signaling [43]. Notably, we observed that high saturated fat intake was associated with increased IL-6 levels (OR = 2.03, 95% CI = 1.00–4.11, p < 0.047), whereas high total fat intake was associated with increased leptin levels (OR = 2.14, 95% CI = 1.12–3.38, p < 0.021) in participants with obesity. These findings support the idea that inflammation and disease risk are influenced by dietary components, particularly fat [44]. A high-fat diet, particularly one rich in saturated fatty acids (SFAs), can trigger a cascade of inflammatory responses in the body. SFAs are known to activate Toll-like receptors (TLRs), specifically TLR4, which is expressed on various immune cells and adipocytes. When SFAs bind to TLR4, it activates the nuclear factor kappa-light-chain-enhancer of activated B cells (NF-κB) pathway, leading to the transcription of pro-inflammatory cytokines [45]. This process results in chronic low-grade inflammation, a hallmark of obesity, characterized by the infiltration of immune cells into adipose tissue.

The positive correlation we observed between leptin levels and energy, carbohydrate, and sugar intakes suggests a potential mechanism for leptin resistance in obesity. The observed weak correlation indicates that only a limited portion of the variance in leptin levels can be

attributed to variations in sugar intake. Consequently, this raises questions about the practical implications of these findings within a clinical context. Previous literature has similarly reported such associations yet emphasized that statistical significance does not equate to meaningful clinical effects. For instance, studies by Sinha et al. [46]. indicate that while certain dietary components may correlate with leptin levels, the clinical significance of these correlations can be limited. Weak correlations may not result in substantial changes in health outcomes or metabolic risk, thereby necessitating a broader evaluation of dietary patterns and lifestyle factors influencing leptin regulation. This finding aligns with how continuous high calorie intake leads to leptin resistance, due to elevated triglyceride and leptin levels impairing leptin transport.

Moreover, our study showed that all anthropometric parameters, except for age and height, were significantly higher in the obese group ($p < 0.001$). Notably, the average waist circumference in the obese group exceeded 90 cm, correlating with increased CVD risk [47] The effects of different fatty acids on inflammation and metabolic parameters are particularly interesting. Consuming an SFA-enriched diet resulted in increased inflammatory marker levels, whereas consuming a MUFA-rich diet caused an increased anti-inflammatory activity, which supports the results of previous studies on the differential impacts of fatty acid types [48] The mechanism may involve the stimulation of peroxisome proliferator-activated receptors, which play regulatory roles in glucose and lipid homeostasis [49]. The elevated leptin levels in participants with obesity, coupled with the association between high fat intake and increased leptin levels, support the concept of leptin resistance in obesity [41] This resistance may be partly caused by impaired leptin transport, potentially influenced by high triglyceride levels, as suggested by previous studies [50].

Although our study showed a trend toward increased leptin levels with high sugar intake, this was not statistically significant. However, animal studies have pointed to fructose as a possible cause of leptin resistance, suggesting that more research is needed to better understand how carbohydrates affect leptin levels in humans [51]. The study demonstrates several strengths, including a large sample size of 384 adults aged 18–59 years, which provides robust statistical power for analyses. It also conducted a comprehensive assessment, including anthropometric measurements, blood biochemistry, inflammatory markers, hormones, and dietary intake. The quantification of specific associations, such as the odds ratio for high saturated fat intake and increased IL-6 levels, adds to the study's rigor. However, the study also has limitations. Its cross-sectional design precludes the establishment of causal relationships between dietary fat consumption and inflammatory markers or leptin levels. The reliance on self-reported dietary intake may introduce recall bias, potentially affecting the accuracy of the dietary data. Additionally, the use of Asian obesity classification criteria may limit the generalizability of findings to non-Asian populations. Moreover, the limitation for this study is the semi-FFQ. Although our semi-FFQ was adapted from a previously validated version for Thai adults with metabolic syndrome risk [29], and underwent expert review for our modifications of additional local foods, the modified version has not yet been specifically validated in our study context, which could affect the precision of dietary intake assessments.

Future research should focus on longitudinal and intervention studies to establish causal relationships between dietary fat consumption, inflammatory markers, and leptin levels in obesity. A closer examination of different types of fats and carbohydrates could elucidate the mechanisms underlying diet-induced leptin resistance. Investigating these associations in diverse populations would broaden the applicability of findings. Examining gene-diet interactions and long-term health outcomes associated with dietary fat-induced changes in inflammatory markers and leptin levels could provide more insights. These directions may lead to more

effective nutritional strategies for obesity management and prevention of related metabolic complications.

## Conclusions

Obesity is a chronic low-grade inflammation. Dietary factors, particularly HFDs, exacerbate inflammation and impact leptin levels, thereby influencing obesity-related metabolic disorders. The results of our study showed a positive correlation between blood biochemistry and TNF-α, IL-6, and leptin levels. Furthermore, energy, carbohydrate, and sugar intakes were positively correlated with leptin levels. Moreover, the results indicated that high fat consumption (SFAs) increased IL-6 and leptin levels in participants with obesity. This relationship may contribute to the development of metabolic syndrome and its associated complications. Our findings highlight the significance of dietary strategies in managing obesity-related health risks and support the role of reduced-fat diets in promoting metabolic health in individuals with obesity.

## Supporting information

**S1 Table. Interaction between blood inflammation and hormonal markers in relation to sugar intake in obese individuals.** Analysis of interactions between inflammatory and hormonal biomarkers stratified by high versus low sugar intake in the obese group (n = 199). (DOCX)

**S2 Table. Interaction between blood inflammation and hormonal markers in relation to saturated fat intake in obese individuals.** Analysis of interactions between inflammatory and hormonal biomarkers stratified by high versus low saturated fat intake in the obese group (n = 199). (DOCX)

**S3 Table. Interaction between blood inflammation and hormonal markers in relation to total fat intake in obese individuals.** Analysis of interactions between inflammatory and hormonal biomarkers stratified by high versus low total fat intake in the obese group (n = 199). (DOCX)

**S1 Data. Raw anthropometric data including fat mass measurements in obese participants.** Individual-level data containing fat mass measurements for the obese study group (n = 199). (XLSX)

## Acknowledgments

We thank the Faculty of Tropical Medicine at Mahidol University for providing the necessary instruments.

## Author Contributions

**Conceptualization:** Pattaneeya Prangthip.

**Data curation:** Sakawrut Poosri.

**Funding acquisition:** Pattaneeya Prangthip.

**Methodology:** Pattaneeya Prangthip.

**Project administration:** Pattaneeya Prangthip.

**Validation:** Sakawrut Poosri, Karani Santhanakrishnan Vimaleswaran, Pattaneeya Prangthip.

**Writing – original draft:** Sakawrut Poosri, Pattaneeya Prangthip.

**Writing – review & editing:** Sakawrut Poosri, Karani Santhanakrishnan Vimaleswaran, Pattaneeya Prangthip.

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
