## [Decision Letter · Decision Letter 0]

17 Oct 2024

PONE-D-24-35820Dietary lipids shape cytokine and leptin profiles in obesity-metabolic syndrome implications: a cross-sectional studyPLOS ONE

Dear Dr. Prangthip

Thank you for submitting your manuscript to PLOS ONE. After careful consideration, we feel that it has merit but does not fully meet PLOS ONE’s publication criteria as it currently stands. Therefore, we invite you to submit a revised version of the manuscript that addresses the points raised during the review process.

Please, pay special attention to the points raised by reviewer 3, regarding methodological issues, such as calculation of the sample, confounders, and analysis of the data.

We look forward to receiving your revised manuscript.

Kind regards,

Víctor Sánchez-Margalet

Academic Editor

PLOS ONE

Journal Requirements: When submitting your revision, we need you to address these additional requirements. 1. Please ensure that your manuscript meets PLOS ONE's style requirements, including those for file naming. The PLOS ONE style templates can be found at https://journals.plos.org/plosone/s/file?id=wjVg/PLOSOne_formatting_sample_main_body.pdf and https://journals.plos.org/plosone/s/file?id=ba62/PLOSOne_formatting_sample_title_authors_affiliations.pdf 2. We suggest you thoroughly copyedit your manuscript for language usage, spelling, and grammar. If you do not know anyone who can help you do this, you may wish to consider employing a professional scientific editing service.  The American Journal Experts (AJE) (https://www.aje.com/) is one such service that has extensive experience helping authors meet PLOS guidelines and can provide language editing, translation, manuscript formatting, and figure formatting to ensure your manuscript meets our submission guidelines. Please note that having the manuscript copyedited by AJE or any other editing services does not guarantee selection for peer review or acceptance for publication.  Upon resubmission, please provide the following: The name of the colleague or the details of the professional service that edited your manuscript A copy of your manuscript showing your changes by either highlighting them or using track changes (uploaded as a *supporting information* file) A clean copy of the edited manuscript (uploaded as the new *manuscript* file)” 3. Thank you for stating the following financial disclosure: "Mahidol University" Please state what role the funders took in the study.  If the funders had no role, please state: ""The funders had no role in study design, data collection and analysis, decision to publish, or preparation of the manuscript.""  If this statement is not correct you must amend it as needed. Please include this amended Role of Funder statement in your cover letter; we will change the online submission form on your behalf. 4. We are unable to open your Supporting Information file "TCTR20240723011". Please kindly revise as necessary and re-upload. 5. We notice that your supplementary tables are included in the manuscript file. Please remove them and upload them with the file type 'Supporting Information'. Please ensure that each Supporting Information file has a legend listed in the manuscript after the references list. 6. Please include captions for your Supporting Information files at the end of your manuscript, and update any in-text citations to match accordingly. Please see our Supporting Information guidelines for more information: http://journals.plos.org/plosone/s/supporting-information.

Reviewers' comments:

Reviewer's Responses to Questions

**Comments to the Author**

1. Is the manuscript technically sound, and do the data support the conclusions?

Reviewer #1: Yes

Reviewer #2: Yes

Reviewer #3: Yes

2. Has the statistical analysis been performed appropriately and rigorously? 

Reviewer #1: Yes

Reviewer #2: Yes

Reviewer #3: No

3. Have the authors made all data underlying the findings in their manuscript fully available?

Reviewer #1: Yes

Reviewer #2: Yes

Reviewer #3: Yes

4. Is the manuscript presented in an intelligible fashion and written in standard English?

Reviewer #1: Yes

Reviewer #2: No

Reviewer #3: Yes

5. Review Comments to the Author

Reviewer #1: This is an interesting study on “Dietary lipids shape cytokine and leptin profiles in obesity-metabolic syndrome implications: a cross-sectional study”. It’s an honour to have the opportunity to review the manuscript.

My comments are appended.

Abstract:

Abstract is written properly; however, I think it is better to elaborate the methodology of anthropometric and biochemical measurements.

Introduction:

Introduction has been written in right direction. However, the author didn’t mention the context of the area and people’s ethnicity. It is okay, if this is the first ever study to explore this in the context of people all over the world, though it is worthy to mention. It is necessary to ensure external validity of any findings.

Methods:

It is not clear how the sample size was calculated.

Is the semi-FFQ previously validated in Thailand?

Results:

The results are written in good order.

Discussion:

Discussion is written properly.

Reviewer #2: The manuscript evaluates the relationship between inflammatory markers, hormones, and dietary fat, linking them to obesity in an Asian population. While the study presents valuable insights, revisions are required to enhance its clarity.

Methods

Line 96: Was there a maximum BMI threshold used as an inclusion criterion?

Lines 105-107: Please, refine the language. The sentence is somewhat confusing. The same for lines 111-112.

Line 117: what is BIA?

Line 138: Is there a reference used for the calculation of HOMA-IR?

Line 142: standardize the citation of reagents/equipment

Line 143: explain the meaning of high-risk values: high risk for metabolic diseases?

Line 149: What time frame was used to assess dietary intake? Even though the FFQ has already been published, it would be helpful to describe it briefly in this manuscript.

Results

Line 166: review the writing.

Line 167: Be specific in the text considering the comparison of obesity between genders, for example, showing the P-value.

Line 175: It would be more effective to begin this section with the final sentence of the paragraph.

Line 464 (table 2): Were the data adjusted for gender? This is important, as men and women exhibit considerable differences in body composition. I suggest showing both analyses.

Line 195: According to figure 1B, HDL level is higher in the non-obese group. Please revise the text to reflect this information accurately.

Figure 1A: Create a graph with the y-axis divided into two segments to highlight the smaller values.

Figure 1A-B: Some of the bars in the graphs appear to be missing SD. Improve the quality of the graphs.

There are 2 legends in figure 1.

Lines 198-205: these data are presented in table 5

Lines 206-208: these data are presented in table 4

Line 202: the reported r value is associated with correlations involving TNF. Specify or provide the corresponding correlation values for leptin in the text.

Table 5: the parameters in the columns (e.g., BL glucose) are somewhat unclear. It would also be more appropriate to include the abbreviations below each table for clarity.

Lines 209-217: starting this topic with the final sentence of the paragraph may make it easier for readers to follow the data.

Line 212: refer to supplementary table 2

Line 213: refer to supplementary table 3

Line 215: the data are presented in table 4, not 5 (leptin versus sugar intake).

Discussion

Line 223: I suggest the authors discuss the higher prevalence of obesity observed among women, considering global data or specific to Asia.

What is the meaning of negative correlations of leptin, IL-6, TNF, and insulin with HDL?

Line 254: reference?

Reviewer #3: This study entitled Dietary lipids shape cytokine and leptin profiles in obesity-metabolic syndrome

implications: a cross-sectional study is interesting. However, I have some comments.

1. "Obesity is a significant risk factor for metabolic syndrome, a cluster of conditions including diabetes, cardiovascular diseases (CVDs), dyslipidemia, and insulin resistance." There isn’t any references for this sentence. Please add reference like these studies: "Association of circulating adipokines with metabolic dyslipidemia in obese versus non-obese individuals"

2. Its better to add some reports about the prevalence rate of obesity across the world based on some meta-analysis or epidemiological studies.

3. The section does a good job of summarizing existing knowledge about obesity and inflammation, mentioning key inflammatory cytokines (TNF-α, IL-6) and leptin. However, it could delve a bit deeper into why these particular markers were chosen for the study. For instance, more emphasis on leptin's role in obesity-related complications could add depth.

4. "Complex factors of genetic susceptibility, sedentary lifestyle, and high-calorie diet consumption comprise the etiology of obesity and metabolic syndrome, resulting in chronic energy imbalance and excessive body fat accumulation" please add references such as 10.1016/j.clinthera.2020.12.021 and 10.1002/ptr.7081

5. How did you calculate sample size for this study, it should be added to the method section.

6. cutoff values for high-risk levels of TNF-α, IL-6, and leptin are provided, which is a strength. However, a brief explanation of why these specific thresholds were chosen (e.g., based on previous research or clinical relevance) would be useful.

7. Statistical Analysis: its better to explain the confounding or covariate used in this study.

8. Were any potential confounders (e.g., age, gender, physical activity) controlled for in the analysis, and how were they selected?

9. the statistical significance of these correlations (e.g., for sugar and leptin levels) should be interpreted more carefully. Although the correlations are statistically significant, are they clinically meaningful? The strength of the correlations (e.g., r-values) could be discussed to provide more context.

10. In the discussion, its better to explain more about mechanism related to observed results.

6. PLOS authors have the option to publish the peer review history of their article (what does this mean?). If published, this will include your full peer review and any attached files.

Reviewer #1: **Yes: **Md Kamruzzaman

Reviewer #2: No

Reviewer #3: No

---

## [Author Response · Author response to Decision Letter 0]

29 Oct 2024

In response to the reviewers' thoughtful comments, we have made substantial revisions to strengthen our manuscript. Key modifications include:

Enhanced methodological clarity by:

Adding detailed sample size calculations

Clarifying the validation status of our semi-FFQ in the Thai context

Providing comprehensive descriptions of anthropometric and biochemical measurement protocols

Standardizing reagent and equipment citations

Strengthened statistical analysis by:

Including gender-adjusted analyses for anthropometric parameters

Adding confounding variable considerations (age, gender, physical activity)

Improving the presentation of correlation analyses

Expanded discussion section by:

Including detailed mechanisms underlying our observations

Adding context regarding obesity prevalence in Asian populations

Providing deeper analysis of the relationship between inflammatory markers and HDL

Enhancing the interpretation of statistical versus clinical significance

All changes are highlighted in the revised manuscript. The work remains original and unpublished, and all authors have approved these revisions. We declare no conflicts of interest.

We thank the reviewers for their valuable input, which has significantly improved our manuscript. We look forward to your evaluation of this revised version.

Sincerely,

Pattaneeya Prangthip

---

## [Decision Letter · Decision Letter 1]

2 Dec 2024

Dietary lipids shape cytokine and leptin profiles in obesity-metabolic syndrome implications: a cross-sectional study

PONE-D-24-35820R1

Dear Dr. Pattaneeya Prangthip

We’re pleased to inform you that your manuscript has been judged scientifically suitable for publication and will be formally accepted for publication once it meets all outstanding technical requirements.

Kind regards,

Víctor Sánchez-Margalet

Academic Editor

PLOS ONE

Additional Editor Comments (optional):

Reviewers' comments:

Reviewer's Responses to Questions

**Comments to the Author**

1. If the authors have adequately addressed your comments raised in a previous round of review and you feel that this manuscript is now acceptable for publication, you may indicate that here to bypass the “Comments to the Author” section, enter your conflict of interest statement in the “Confidential to Editor” section, and submit your "Accept" recommendation.

Reviewer #1: All comments have been addressed

Reviewer #2: All comments have been addressed

Reviewer #3: All comments have been addressed

2. Is the manuscript technically sound, and do the data support the conclusions?

Reviewer #1: Yes

Reviewer #2: Yes

Reviewer #3: Yes

3. Has the statistical analysis been performed appropriately and rigorously? 

Reviewer #1: Yes

Reviewer #2: Yes

Reviewer #3: Yes

4. Have the authors made all data underlying the findings in their manuscript fully available?

Reviewer #1: Yes

Reviewer #2: Yes

Reviewer #3: Yes

5. Is the manuscript presented in an intelligible fashion and written in standard English?

Reviewer #1: Yes

Reviewer #2: Yes

Reviewer #3: Yes

6. Review Comments to the Author

Reviewer #1: Thanks for addressing issues raised during the review process. Quality of the manuscript has now been increased and suitable for publication.

Reviewer #2: (No Response)

Reviewer #3: The researchers have been able to correct the comments raised in the previous version of the manuscript and the manuscript in its current form meets the necessary conditions for acceptance.

7. PLOS authors have the option to publish the peer review history of their article (what does this mean?). If published, this will include your full peer review and any attached files.

Reviewer #1: **Yes: **Md Kamruzzaman

Reviewer #2: No

Reviewer #3: No

---

## [Editor Report · Acceptance letter]

10 Dec 2024

PONE-D-24-35820R1 

PLOS ONE

Dear Dr. Prangthip, 

I'm pleased to inform you that your manuscript has been deemed suitable for publication in PLOS ONE. Congratulations! Your manuscript is now being handed over to our production team.

Kind regards, 

on behalf of

Dr. Víctor Sánchez-Margalet 

Academic Editor

PLOS ONE